# CESCR: CP-ABE for efficient and secure sharing of data in collaborative ehealth with revocation and no dummy attribute

**Kennedy Edemacu[1], Beakcheol Jang[2]\*, Jong Wook Kim[1]\***

**1** Department of Computer Science, Sangmyung University, Seoul, South Korea, **2** Graduate School of Information, Yonsei University, Seoul, South Korea

\* jkim@smu.ac.kr (JWK); bjang@yonsei.ac.kr (BJ)

**Data Availability Statement:** All relevant data are within the manuscript.

**Funding:** This research was supported by the Basic Science Research Program through the National

## Abstract

With the rapid advancement of information and communication technologies, there is a growing transformation of healthcare systems. A patient's health data can now be centrally stored in the cloud and be shared with multiple healthcare stakeholders, enabling the patient to be collaboratively treated by more than one healthcare institution. However, several issues, including data security and privacy concerns still remain unresolved. Ciphertext-policy attribute-based encryption (CP-ABE) has shown promising potential in providing data security and privacy in cloud-based systems. Nevertheless, the conventional CP-ABE scheme is inadequate for direct adoption in a collaborative ehealth system. For one, its expressiveness is limited as it is based on a monotonic access structure. Second, it lacks an attribute/user revocation mechanism. Third, the computational burden on both the data owner and data users is linear with the number of attributes in the ciphertext. To address these inadequacies, we propose CESCR, a CP-ABE for efficient and secure sharing of health data in collaborative ehealth systems with immediate and efficient attribute/user revocation. The CESCR scheme is unbounded, i.e., it does not bind the size of the attribute universe to the security parameter, it is based on the expressive and non-restrictive ordered binary decision diagram (OBDD) access structure, and it securely outsources the computationally demanding attribute operations of both encryption and decryption processes without requiring a dummy attribute. Security analysis shows that the CESCR scheme is secure in the selective model. Simulation and performance comparisons with related schemes also demonstrate that the CESCR scheme is expressive and efficient.

## 1 Introduction

Collaborative ehealth is a paradigm that enables sharing of electronic health information between healthcare stakeholders for efficient coordination and quality healthcare delivery to patients. In modern healthcare systems, the paradigm is playing a vital role in patients being simultaneously treated by multiple healthcare institutions [1]. In collaborative ehealth systems, the electronic health information can be obtained through wearable and embeddable health

Research Foundation of Korea (NRF-2020R1F1A1072622).

**Competing interests:** The authors have declared that no competing interests exist.

sensors [2, 3], medical recordings from health facilities, etc., and be outsourced to the cloud for sharing [4–6]. For example, consider a patient being treated simultaneously by two hospitals H-A and H-B for a heart problem and diabetes, respectively. As part of the treatment plan, H-A gives the patient a wearable health sensor to monitor her daily heart rate. Through a mobile device, the health sensor data is outsourced to the cloud for access by both H-A and H-B. This way, the need for repeated and duplicated medical examinations by H-B is minimized.

As fascinating as it may be, there are still several concerns that need to be addressed for its total acceptance. In particular, the use of third party servers for data storage presents privacy and security issues which are increasingly becoming the biggest concern in collaborative ehealth systems. Adoption of the traditional access control techniques can be used to address the data privacy and security concern in collaborative ehealth. However, these techniques only allow coarse-grained access policies which are not ideal for scalable environments.

An attractive solution is to adopt the attribute-based encryption (ABE) scheme which allows for the realization of fine-grained access policies [7]. ABE is primarily divided into: key-policy attribute-based encryption (KP-ABE) [7, 8] and ciphertext-policy attribute-based encryption (CP-ABE) [9] which is our focus in this work. In CP-ABE, the ciphertext is associated with an access policy and the user key is labeled with a set of attributes. Since its inception, CP-ABE has attracted a lot of attention for fine-grained access control in cloud environments. In [10–16], different CP-ABE schemes are proposed for fine-grained access control of data in the cloud. However, the schemes rely on access structures that are either monotonic or restrictive, thus affecting the expressiveness and efficiency of the resulting schemes. As a result, ordered binary decision diagram (OBDD) access structure has been proposed and used for construction of expressive and efficient CP-ABE schemes in [17, 18].

Although the traditional OBDD-based CP-ABE schemes are expressive, their direct adoption for collaborative ehealth does not seem suitable. It is still necessary to simultaneously resolve the issues of unboundedness, expressiveness, efficiency and attribute/user revocation to ensure their usability and effectiveness for fine-grained access control in collaborative ehealth environments.

## Attribute/user revocation and collusion resistance

Revoking misbehaving/compromised or obsolete users is a key requirement in collaborative ehealth systems [19]. However, the users share attributes and revoking a user of an attribute affects other users bearing the same attribute. As such, techniques like the expiration times [20, 21], version numbers [22, 23], attribute groups [24, 25], etc., have been proposed to achieve attribute/user revocations in systems deploying ABE schemes. The most important aspect in revocation is that collusion between revoked and non-revoked users should be prevented.

## Unboundedness

ABE schemes are alternatively classified into "bounded" and "unbounded" schemes. In "bounded" schemes, the total number of attributes in the attribute space is fixed during setup and is polynomially bounded in the security parameter. The bounding of the size of the attribute universe can have undesirable effects on systems deploying ABE schemes. A smaller bound might result in the system exhaustion and a need for complete rebuilding when expansion is required. For example, consider the previous scenario in which the patient suffering from the heart disease is being treated by a doctor in hospital H-A. In a smaller bound ABE scheme deployment, the attribute universe leveraged for encryption and user key generation

can be set as {*hospital*, *department*, *profession*}. However, at a later time, if the patient requires her data to be accessed only by experienced doctors, a new attribute "*experience*" might be introduced. In this bounded setting, to generate parameters associated with the "*experience*" attribute, the system will have to be completely rebuilt and additional expenses are incurred to re-encrypt all the ciphertexts. On the other hand, a larger bound might result in inefficient use of system resources as some parameters might be redundantly stored. Meanwhile, in the "unbounded" schemes, the total number of attributes in the attribute space is not bounded during setup and can expand exponentially.

## Efficiency

In collaborative ehealth, several less powerful computing devices are involved. Consider the same scenario in which the patient suffering from the heart disease is being given a sensor device to monitor her daily activities by H-A. The captured sensor data is encrypted and sent to the cloud for analysis and diagnosis by doctors in H-A. In such a setting, the patient might be mobile and most likely use her mobile phone which has limited computing power to perform the data encryption before sending it to the cloud. This necessitates outsourcing of the computationally demanding ABE attribute operations incurred during encryption to the cloud. The same might apply to the doctor and thus, necessitates outsourcing of computationally demanding attribute operations incurred during decryption to the cloud. The most common technique used for secure outsourcing of computations in ABE involves the use of a dummy attribute which is borne by all the users in the system [26].

## Expressiveness

Apart from the mentioned issues, expressiveness is another important issue to consider in attribute-based access control schemes. Several existing schemes support restrictive and monotonic access structures which are less expressive. A more expressive and non-restrictive access structure is the OBDD access structure and it can represent any non-monotonic boolean formula.

 **Our contribution.** In this study, we address the security and privacy concerns in collaborative ehealth by proposing CESCR scheme. In CESCR, we simultaneously address the issues of attribute/user revocation, user collusion, unboundedness, expressiveness and efficiency. We provide a comprehensive security analysis, and simulation and performance evaluation for the CESCR scheme. The security analysis, and the simulation and performance evaluation results show that CESCR is secure and efficient for sharing of health data in collaborative ehealth systems. Specifically, CESCR scheme has the following features:

- *Attribute/user revocation*: In CESCR, we adapt the attribute group approach [24]. Attribute groups are created whose members are users sharing the same attribute. A user can belong to multiple attribute groups depending on the number of attributes he/she bears. Each attribute group has a unique key only known to its group members. When a user is revoked of an attribute, a new attribute group is generated and broadcast to all the group members except the revoked user and the ciphertext element associated with the revoked attribute is updated. Unlike in [24, 25], in CESCR, the attribute keys are tightly and efficiently bound to the user identity which helps to prevent collusion attacks.

- *Unboundedness*: In CESCR, the size of attribute universe is not bounded to the security parameter and thus, the number of attributes can expand exponentially while keeping the number of system public parameters constant. To achieve this, we propose a novel technique

in which the only attribute elements in CESCR's ciphertexts are those associated with the attribute groups of the ciphertext attributes.

- *Efficiency*: CESCR securely outsources the computationally demanding attribute operations in both encryption and decryption to the cloud. But unlike other schemes that leverage dummy attributes to achieve secure outsourcing, the CESCR scheme does not require a dummy attribute.

- *Expressiveness*: CESCR uses the OBDD access structure, which is non-monotonic and non-restrictive. Thus, it can handle any non-monotonic access policy expressable using the OBDD access structure.

- *User collusion resistance*: In CESCR, the decryption keys are bound to the user identity, which makes it collusion resistant.

**Paper organization.** The rest of the paper is organized as follows, in Section 2, we present the related works. In Section 3, we present the summary of access structure, and mathematical and cryptographic complexity assumptions used in this work. Section 4 covers the system architecture, the formal scheme definition and the security model. In Section 5, we present the concrete construction of the CESCR scheme. We present the security analysis of our scheme in Section 6. Sections 7 and 8 present the simulation and performance evaluation, and conclusion, respectively.

## 2 Related work

The demand for improved healthcare service delivery is constantly increasing. Additionally, healthcare services are shifting from treatment oriented to proactive prevention. To achieve this, there is a need to have electronic health information centrally stored to be accessed and shared with healthcare stakeholders. For this reason, cloud-based health systems have turned out to be useful. In [27], an intelligent cloud-based healthcare service system is designed in which health sensors are utilized to obtain health data from a patient and sent to the cloud for storage and analysis. The system provides real time monitoring of patients for chronic diseases. In [28], Miah *et al.* designed a cloud-based ehealth system to enable health workers to collaborate for identifying and treating non-communicable diseases in rural areas of developing countries. In their system, less knowledgeable health workers in rural communities record health information from patients which are then stored in the cloud and made accessible to remotely located but knowledgeable doctors for analysis and recommendations. [29, 30] proposed integration of smart homes in cloud-based health systems. Their proposed system utilizes the smart home environment to gather health information which is then sent to the cloud for analysis.

Although the above-discussed studies have proposed and designed interesting health systems, none of them has focused on the data security and privacy issues encountered during health data sharing. To address the above issues, [31] designed a scheme that provides location privacy for patients and doctors in IoT-based health systems. The scheme employs the Chinese remainder theorem to preserve location privacy. Similarly, in [32], Azees *et al.* proposed schemes for anonymous authentication of patients and doctors in IoT-based health systems, and preserve the confidentiality of health data exchanged between the entities. [5, 6, 21, 33, 34] have studied and proposed ABE schemes for secure sharing of electronic health information in cloud-based health systems. ABE was originally proposed by Sahai and Waters in the form of fuzzy identity-based encryption [7]. It has since then been categorized as: KP-ABE in which

secret keys are associated with access policies while ciphertexts are associated with attribute sets [8], and CP-ABE in which secret keys are associated with attribute sets while ciphertexts are associated with access policies [9]. Cheung and Newport then proposed a CP-ABE scheme based on the AND-gate access structure [11]. In the same work, they presented a security proof for their scheme in the standard model. Further ABE schemes have been proposed focusing on multi-authority [35, 36], hidden access-structure [37, 38] and hierarchy [39, 40]. However, these schemes rely on access structures that are either monotonic or restrictive. [17, 18] proposed CP-ABE schemes based on the non-monotonic and non-restrictive OBDD access structure. However, their schemes are bounded and aggregate attribute elements in ciphertext and decryption keys together, which makes it difficult to integrate an efficient and immediate attribute/user revocation.

A number of attribute/user revocation approaches have been proposed for ABE systems. In [20, 21, 41], a revocation list is included during encryption which is updated periodically. A user whose ID is listed in the revocation list is denied key updates and thus unable to decrypt the updated ciphertext. One drawback with this approach is that, revocations are not immediate. [24, 25, 42] proposed attribute group approach, in which attribute groups whose members are users sharing the same attribute are created. Each group is assigned a key only known to its members. Whenever a user is revoked from the group, a new key is generated and broadcast to all the group members except the revoked user. However, the [24] scheme suffers from collusion attacks, the [25] scheme is computationally inefficient and the [42] scheme is less expressive as it relies on the monotonic LSSS access structure. Version number approach is proposed in [22, 43]. In these schemes, user keys and ciphertexts are assigned version numbers, whenever a user is revoked of an attribute, an update key is generated and forwarded to all non-revoked users and their key version number is increased by one. The ciphertext is also updated and its version number gets increased by one. Further ABE schemes focusing on efficiency through generation of fixed-sized ciphertexts and outsourcing are presented in [26, 44, 45]. In [26], to securely outsource computations to the cloud, an inefficient approach in which a redundant dummy attribute which is shared by all the users is used. The elements associated with the dummy attribute are never updated.

The first construction of an unbounded (large universe) KP-ABE scheme was given by [46] in the composite order groups. Rouselakis and Waters in [47] constructed unbounded KP-ABE and CP-ABE schemes supporting LSSS access structures in the prime order groups. The construction in [47] was used by [48] to construct an unbounded CP-ABE scheme with partially hidden LSSS access structures in prime order groups. Recently, Zhang *et al.* [49] proposed an unbounded CP-ABE scheme for security and privacy protection in smart health systems. Their scheme partially hides LSSS access structures and its construction is based on the composite order groups. An unbounded CP-ABE scheme based on prime order group that supports partially hidden AND access structures is proposed in [50]. A large universe CP-ABE scheme supporting traceability and revocation is proposed in [51]. However, the scheme supports only the monotonic LSSS access structures and leverages the direct revocation mechanism in which the revocation lists are included during encryption. As such it is less expressive and does not achieve immediate attribute/user revocation.

In this work, we adapt the attribute group approach of [24, 25, 42] to achieve immediate and efficient attribute/user revocations. However, unlike in previous works, to prevent collusion attacks, the attribute group keys are efficiently bound to the user identities in this work. The unboundedness in our scheme is achieved through a novel technique that limits the attribute elements in the ciphertexts to only those associated with the attribute group keys of the ciphertext attributes. Our scheme also securely outsources computations to the cloud with no need for a redundant dummy attribute. To achieve expressiveness, we leverage the OBDD

access structure. However, unlike in [17, 18], the attribute elements in the ciphertext and secret keys are not bound together, thus making it possible to achieve efficient and immediate attribute/user revocations.

# 3 Preliminaries

In this section, we present the summaries of bilinear map, complexity assumption, access structure, and the CP-ABE scheme that lays the foundation for the construction of the CESCR scheme.

## 3.1 Bilinear map

As in [9], let $\mathbb{G}$ and $\mathbb{G}_T$ be two cyclic multiplicative groups of prime order $p$ and $g$ be the generator of $\mathbb{G}$. A bilinear map is defined as, $e : \mathbb{G} \times \mathbb{G} \to \mathbb{G}_T$, subject to satisfaction of the following properties:

1.  Bilinearity. That is, $e(u^x, v^y) = e(u^y, v^x) = e(u, v)^{xy}$ for a given $u, v \in \mathbb{G}$ and $x, y \in \mathbb{Z}_p$.

2.  Non-degeneracy. That is, $\exists\, u, v \in \mathbb{G}$ such that $e(u, v) \neq 1$.

3.  Computability. That is, $\forall\, u, v \in \mathbb{G}$, $e(u, v)$ is computationally feasible.

## 3.2 Decisional Bilinear Diffie-Hellman (DBDH) assumption

*Definition 1*: The DBDH [14] assumption states that, given two tuples $(g, g^a, g^b, g^c, e(g, g)^{abc})$ and $(g, g^a, g^b, g^c, e(g, g)^z)$, where $a, b, c, z \in_R \mathbb{Z}_p$, a probabilistic polynomial time algorithm $\mathcal{B}$ that outputs $\{0, 1\}$ can distinguish the two tuples with at most a negligible advantage $\varepsilon$, i.e., $|Pr[\mathcal{B}(g, g^a, g^b, g^c, e(g, g)^{abc}) = 0] - Pr[\mathcal{B}(g, g^a, g^b, g^c, e(g, g)^z) = 0]| \leq \varepsilon$.

## 3.3 Access structure

*Definition 2*: An access structure is a rule $\mathbb{R}$ that returns 1 if an attribute set $S$ satisfies $\mathbb{R}$ ($S \vDash \mathbb{R}$). Otherwise it returns 0. In this work, the access structure used is the ordered binary decision diagram (OBDD) access structure which is non-monotonic and non-restrictive.

## 3.4 OBDD access structure

*Definition 3*: An OBDD access structure is a rooted, directed acyclic graph ($G = (V, E)$) for a boolean function $f(a_0, \cdots, a_n)$ over a set of boolean variables $\{a_0, \cdots, a_n\}$ with a pre-defined variable ordering [52]. Where the boolean variables depict the attributes and $n$ is the number of attributes in the set. The graph has the following properties:

1.  There are two kinds of nodes in the graph $G$, i.e., $V$ is either a terminal or a non-terminal node.

2.  Each non-terminal node in $G$ has two child nodes $low(v)$ and $high(v)$. Also, each non-terminal node is labeled with a 4-element tuple $(i, id, low(v), high(v))$, where $i \in I$ is the serial number of the attribute represented by the node, $id \in ID$ is a unique number assigned for the identification of the node, and $low(v) \in V$ and $high(v) \in V$ are the serial numbers of the node's $low(v)$ and $high(v)$ child nodes, respectively. $I$ is the set of attributes in the access structure and $ID$ is the node identity universe.

3.  There are two terminal nodes labeled as 1 and 0, and they neither represent an attribute nor have child nodes.

4. Each variable (attribute) appears only once along a directed path from the root node to a child node.

5. There are no identical non-terminal nodes, i.e., non-terminal nodes should not share the same *id*, *low(v)* and *high(v)* elements.

6. No node has identical *low(v)* and *high(v)* nodes, i.e., $low(v) \neq high(v)$.

**OBDD access structure satisfaction.**   OBDD access structure satisfaction process is done recursively. Given an attribute set *S*, starting from the root node, *S* is compared with the attribute value stored in the node. If an element in *S* matches the current node's attribute, *S* is forwarded to the *high(v)* child node. Otherwise, it is forwarded to the *low(v)* child node. This is done repeatedly until it is either forwarded to the 1 terminal node or the 0 terminal node. If the 1 terminal node is reached at the end of the process, *S* satisfies the OBDD access structure. Otherwise, *S* does not satisfy the OBDD access structure.

As an example, consider an access policy represented by the following boolean function $f(a_0, a_1, a_2) = a_0.a_1 + a_0.a_2 + a_1.a_2$. The OBDD access structure depicting the described access policy is shown in Fig 1. All the paths from the root node to the 1 terminal node satisfy the OBDD access structure. Thus, the paths, $a_0 a_1$, $\bar{a}_0 a_1 a_2$ and $a_0 \bar{a}_1 a_2$ satisfy the OBDD access structure. However, the paths, $\bar{a}_0 \bar{a}_1$, $\bar{a}_0 a_1 \bar{a}_2$ and $a_0 \bar{a}_1 \bar{a}_2$ do not satisfy the OBDD access structure as they lead to the 0 terminal node.

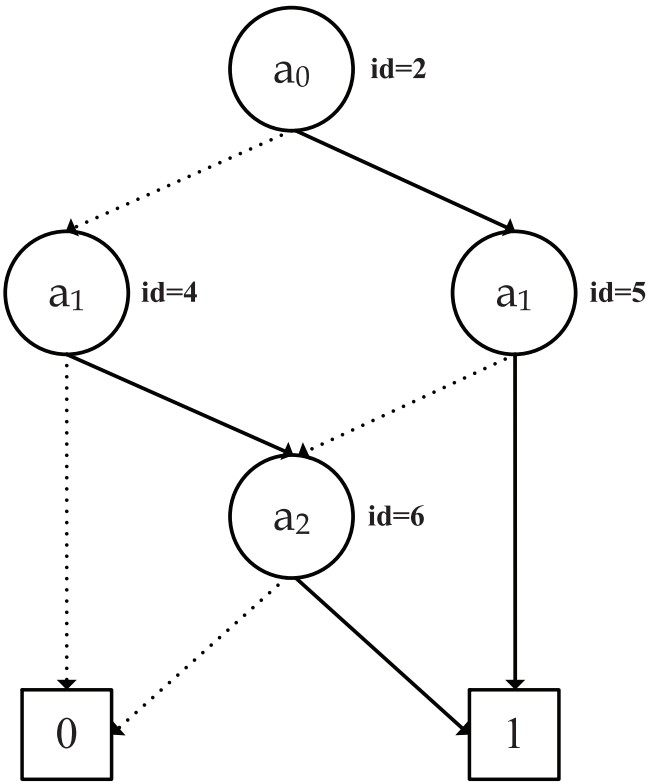

**Fig 1.  An OBDD access structure depicting the $f(a_0, a_1, a_2) = a_0.a_1 + a_0.a_2 + a_1.a_2$ access formula with variable ordering as: $a_0 < a_1 < a_2$.** The solid arrows represent the edges leading to the nodes' *high(v)* child nodes and the dotted arrows represent the edges leading to the nodes' *low(v)* child nodes.

### 3.5 Review of the CP-ABE scheme based on the OBDD access structure

In this section, we present the summary of the conventional CP-ABE scheme [17] based on the OBDD access structure that lays the foundation for the construction of our proposed CESCR scheme and proceeds as follows:

1.  Setup($\lambda$)$\rightarrow$($pp$, $mk$): the algorithm chooses the groups and defines the bilinear map as defined in the Section 1. It then randomly chooses $y\in_R\mathbb{Z}_p$ and computes $Y = e(g, g)^y$. For each attribute in the universe, it randomly chooses $t_i\in_R\mathbb{Z}_p$ and computes $T_i = g^{t_i}|_{i\in\mathcal{U}}$, where $\mathcal{U}$ is the attribute universe. It publishes the public parameters $pp$ as: $(e, g, G, Y, T_i|_{i\in\mathcal{U}})$ and the master key $mk$ as: $(y, t_i|_{i\in\mathcal{U}})$.

2.  KeyGen($S$, $mk$)$\rightarrow$($sk$): It computes the secret key $sk$ associated with the attribute set $S$. It first randomly chooses $r\in_R\mathbb{Z}_p$ and computes $D = g^{y-r}$ and $D_i = g^{(r/\sum_{i\in S}t_i)}$. The secret key $sk$ is $(D, D_i)$.

3.  Encrypt($M$, $pp$)$\rightarrow$($CT$): The data owner first defines an OBDD access structure. The Encrypt algorithm then randomly chooses $s\in_R\mathbb{Z}_p$ and generates the ciphertext $CT$ as: $(OBDD, C_1 = M.Y^s, C_2 = g^s, C_{R_t} = g^{(\sum_{i\in I}t_i.s)}|_{R_t\in R})$. Where $I$ is the attribute set in the OBDD access structure and $R$ is the set of paths that satisfy the OBDD access structure.

4.  Decrypt($CT$, $sk$)$\rightarrow$$M/\perp$: If the user attribute set $S$ satisfies the OBDD access structure, the algorithm computes, $e(C_2, D).e(C_{R_t}, D_i) = e(g, g)^{s.(y-r)}.e(g, g)^{s.r} = e(g, g)^{y.s} = Y^s$. The user then recovers $M$ by computing $C_1/Y^s$. Otherwise, the algorithm returns $\perp$.

## 4 System architecture, formal definition and security model

In this section, we present the system architecture, the formal definition of the CESCR scheme and the security model.

### 4.1 System architecture

Shown in Fig 2 is the system architecture depicting the main entities in our scheme which are described as follows:

**Trusted Authority (TA)**. The TA is a trusted entity that is in-charge of the system initialization, and it also authorizes the data users and the data owner. The TA initializes the system by generating the system public parameters which are made available to all the other entities, and the master key which is kept secret. It authorizes data users through issuing keys associated with user attribute sets. If necessary, the TA also issues a key to the data owner. Additionally, the TA generates attribute group information which it shares with the cloud. We assume the TA is mostly online.

**Data Owner (DO)**. The DO is an entity that owns and manages the outsourced data in the form of ciphertexts. The DO can be a patient or a hospital responsible for managing the patient's data. The outsourced data can be medical recordings obtained from a hospital or health data obtained from health sensors attached to the patient. The DO has either a local server or a smart device that is used to perform partial encryption tasks. Before outsourcing the health data, the DO defines an access policy which is securely sent together with the partially encrypted data to the cloud.

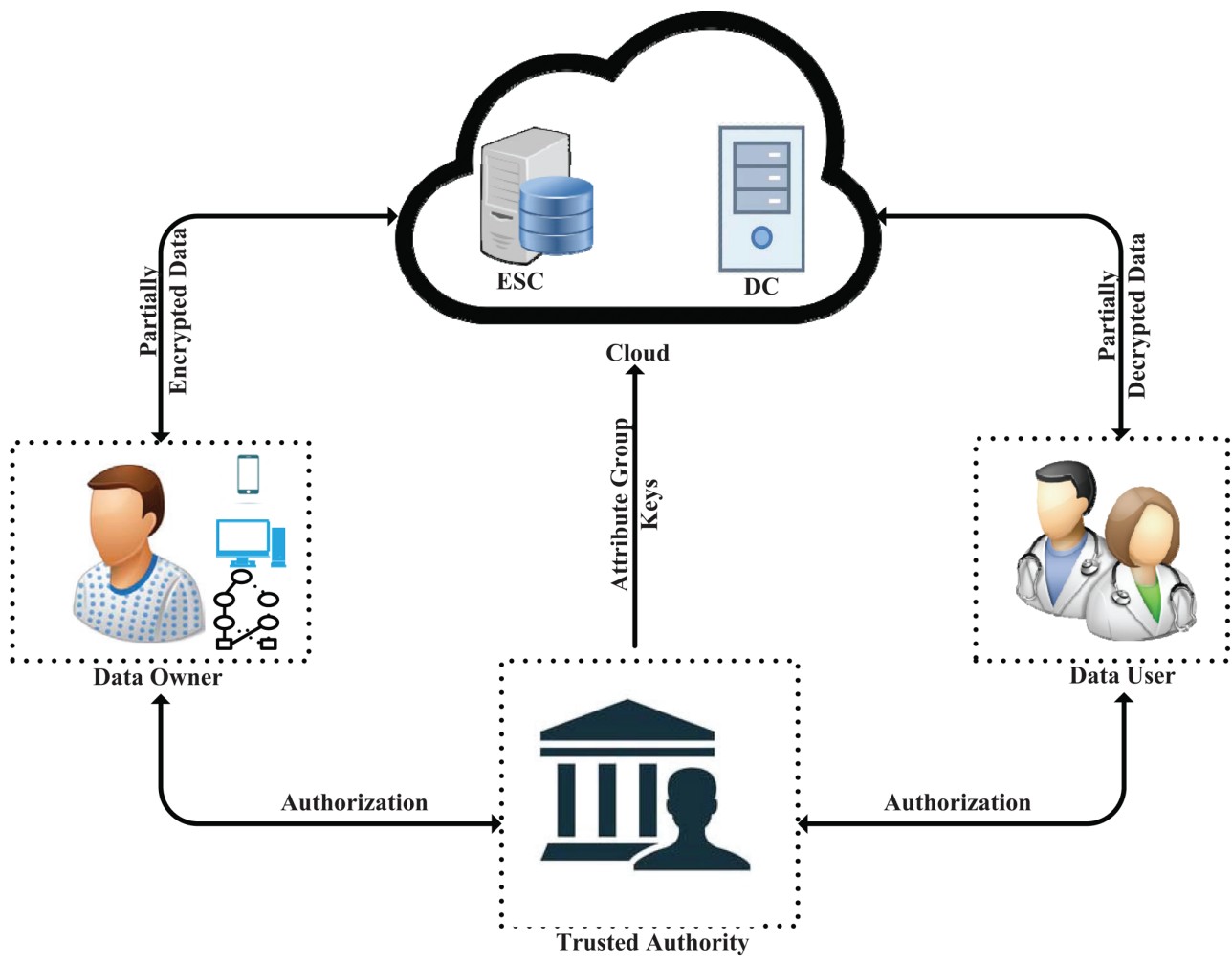

**Fig 2. An architecture of our scheme depicting the entities involved.**

**Data User (DU).** The DU is an entity that uses the patient's data. Doctors, researchers, pharmacists, etc., are some of the examples of DU. Each DU has a set of attributes and attribute associated keys. If the DU's attribute set satisfies the access policy embedded in the ciphertext, he/she can successfully decrypt the ciphertext and use the patient's data. Otherwise, the decryption fails.

**Cloud.** The cloud is an entity that stores and performs partial computations on the health data. In this work, we categorize the cloud into two: the encryption and storage cloud (ESC) and the decryption cloud (DC). The ESC receives the partially encrypted data from the DO, completes the generation of the ciphertext and stores it for sharing with the DUs. Meanwhile, the DC securely receives attribute associated keys from a DU and ciphertext from the ESC to perform partial decryption. We assume the cloud is honest but curious.

## 4.2 Formal definition of CESCR

A CESCR scheme consists of ten algorithms which are described as follows:

- Setup($1^\lambda$)→($pp$, $mk$): The Setup algorithm is executed by the TA. It takes as input the security parameter $\lambda$ and generates the public parameters $pp$ and the master key $mk$ as its output.

- KeyGen($S$, $uid$, $mk$, $pp$)→($D_{i1}$, $D_{i2}$): The KeyGen algorithm is executed by the TA. It takes the public parameters $pp$, the master key $mk$, a user identity $uid$ and a set of attributes $S$ as inputs. It generates the decryption keys ($D_{i1}$, $D_{i2}$) associated with the attributes in $S$ as its output.

- KEKGen($i$, $k_i$, $v_j$, $uid$, $mk$, $pp$)→($KEK_i$): The KEKGen algorithm is executed by the TA. The algorithm takes the public parameters $pp$, the master key $mk$, a user identity $uid$, a minimum cover node $v_j$, an attribute group key $k_i$ and an attribute $i$ as its inputs. It outputs a key encryption key (KEK) associated with the attribute $i$.

- Encrypt($OBDD$, $M$, $pp$)→($CT_l$): The Encrypt algorithm is executed by the DO. The algorithm takes the DO defined access policy $OBDD$, the data to be encrypted $M$ and the public parameters $pp$ as its inputs. It generates a partial ciphertext $CT_l$ as its output.

- CldEncrypt($CT_l$, $k_i$, $v_j$, $pp$)→($CT$): The CldEncrypt is executed by the ESC. It takes as input the public parameters $pp$, the partial ciphertext $CT_l$, attribute group keys $k_i$(s) and the minimum cover nodes $v_j$ associated with each attribute in the access structure, and generates a complete ciphertext $CT$ as its output.

- CldDecrypt($CT$, $D_{i1}$, $D_{i2}$, $KEK_i$, $pp$)→($C_{tkn}$/⊥): The CldDecrypt algorithm is executed by the DC. The algorithm takes as input the public parameters $pp$, a DU's decryption key elements $D_{i1}$ and $D_{i2}$, a DU's key encryption key $KEK_i$ and a ciphertext $CT$. If the set of the DU's attributes satisfy the access structure $OBDD$, the algorithm generates a token $C_{tkn}$ as its output. Otherwise, it generates ⊥.

- Decrypt($C_{tkn}$, $CT$, $pp$)→($M$): The Decrypt algorithm is executed by the DU. It takes the public parameter $pp$, the ciphertext $CT$ and the token $C_{tkn}$ as its inputs. It recovers $M$ as its output.

- UpInfo($i$, $pp$)→($uk_i$): The UpInfo algorithm is executed by the TA after an attribute revocation. It takes as input the public parameters $pp$ and a revoked attribute $i$. The algorithm outputs an update key $uk_i$ for the revoked attribute $i$.

- CTUpdate($CT$, $uk_i$, $i$, $pp$)→($CT'$): The CTUpdate algorithm is executed by the ESC after an attribute revocation. It takes the public parameters $pp$, the revoked attribute $i$, an update key $uk_i$ and the ciphertext $CT$ as its inputs. It outputs an updated ciphertext $CT'$.

- KeyUpdate($i$, $uk_i$, $KEK_i$, $pp$)→($KEK_i'$): The KeyUpdate algorithm is executed by the DU who bears a revoked attribute $i$. The algorithm takes the revoked attribute $i$, an update key $uk_i$, a key encryption key $KEK_i$ and the public parameters $pp$ as its inputs. It outputs an updated key encryption key $KEK_i'$ associated with the revoked attribute $i$.

### 4.3 Security model

In this subsection, we give a security model for the CESCR scheme. The security model is described as a CPA game played between a probabilistic polynomial time (PPT) adversary $\mathcal{A}$ and a challenger, and proceeds as follows:

- **Init**: The adversary $\mathcal{A}$ declares a challenge access structure $\mathcal{OBDD}^*$ and an attribute $i^*$ to the challenger.

- **Setup**: The challenger runs the ($pp$, $mk$)←Setup($1^\lambda$) algorithm. The challenger forwards the public parameters $pp$ to the adversary $\mathcal{A}$ and keeps the master key $mk$.

- **Phase 1**: The adversary $\mathcal{A}$ issues polynomially bounded series of key queries by each time submitting an attribute set $S$ and a user identity $uid$ to the challenger. $S$ satisfies the challenge access structure $\mathcal{OBDD}^*$ but the attribute $i^*$ is revoked. The challenger executes the $(D_{i1},$ $D_{i2})$←KeyGen($S$, $uid$, $mk$, $pp$) and $KEK_i$←KEKGen($i$, $k_i$, $v_j$, $uid$, $mk$, $pp$) algorithms, and gives $D_{i1}$, $D_{i2}$ and $KEK_i$ to adversary $\mathcal{A}$. The adversary $\mathcal{A}$ may also decide to ask for update key for an attribute $i \neq i^*$. The challenger executes the $uk_{i \neq i^*}$←UpInfo($i$, $pp$) algorithm and sends to $\mathcal{A}$ the update key $uk_{i \neq i^*}$.

- **Challenge**: Once the adversary $\mathcal{A}$ decides that Phase 1 is over, it submits two messages $M_0$ and $M_1$ of equal lengths to the challenger and sets $\mathcal{OBDD}^*$ as the access structure and $i^*$ as the revoked attribute. The challenger flips a coin $\mu \in \{0, 1\}$ and encrypts $M_\mu$ by executing the $CT_I$←Encrypt($\mathcal{OBDD}^*$, $M_\mu$, $pp$) algorithm. The challenger then completes the encryption by running the $CT$←CldEncrypt($CT_I$, $k_i$, $v_j$, $pp$) algorithm to generate the ciphertext $CT$. The challenger further updates the ciphertext by executing the $CT'$←CTUpdate($CT$, $uk_{i^*}$, $i^*$, $pp$) algorithm to generate $CT'$. The challenger then sends to $\mathcal{A}$ the $CT'$ as its challenge ciphertext.

- **Phase 2**: The adversary $\mathcal{A}$ continues to adaptively issue key queries to the challenger with the restriction that the submitted attribute sets satisfy the $\mathcal{OBDD}^*$ access structure but $i^*$ is revoked.

- **Guess**: $\mathcal{A}$ then outputs a guess $\mu' \in \{0, 1\}$. The adversary $\mathcal{A}$ wins the game if $\mu = \mu'$. $\mathcal{A}$ wins the game with an advantage defined as $|Pr[\mu' = \mu] - \frac{1}{2}|$.

*Definition 4*: A CP-ABE scheme with attribute revocation, and outsourced encryption and decryption is selective secure if all PPT adversaries have at most a negligible advantage in winning the defined CPA security game.

## 5 CESCR scheme construction

In this section, we present a concrete construction of the CESCR scheme. The construction is divided into five phases and proceed as follows:

1. **Setup**
   The setup phase initializes the system through the Setup algorithm. Let $\mathbb{G}$ and $\mathbb{G}_T$ be two cyclic multiplicative groups of prime order $p$, $g$ be the generator of $\mathbb{G}$, and $e : \mathbb{G} \times \mathbb{G} \to \mathbb{G}_T$ be a bilinear map as defined in Section 3. A hash function $H : \{0, 1\}^* \to \mathbb{G}$ is also defined. Let the attribute universe be $\mathcal{U}$.
   $Setup(1^\lambda) \to (pp, mk)$: The setup algorithm randomly chooses $y, \alpha \in_R \mathbb{Z}_p$. It then computes $h_1 = g^{1/\alpha}$, $h_2 = g^\alpha$ and defines $Y = e(g, g)^y$. It publishes the public parameters $pp$ as, $pp = (e, g, h_1, h_2, \mathbb{G}, Y)$ and keeps the secret master key $mk$ as, $mk = (\alpha, y)$.

2. **Key generation**
   The key generation phase comprises two algorithms: KeyGen and KEKGen algorithms which are both executed by the TA.
   $KeyGen(S, uid, mk, pp) \to (D_{i1}, D_{i2})$: The KeyGen algorithm generates the user secret key $(D_{i1}, D_{i2})$. To generate the secret key for a user $uid$ with attribute set $S = \{a_1, a_2, \cdots, a_n\}$, where $n$ is the number of attributes in $S$, the algorithm first randomly chooses $z_1, z_2, \cdots, z_{n-1} \in_R \mathbb{Z}_p$ and computes $z_n$ as $y - \sum_{i=1}^{n-1} z_i \bmod p$. Also for each attribute in $S$, the algorithm randomly chooses $r_i \in_R \mathbb{Z}_p$. It then computes the user secret key $(D_{i1}, D_{i2})$ with respect to the attribute set $S$ as:

$$D_{i1} = g^{z_i}.H(uid)^{r_i}, D_{i2} = H(uid)^{\alpha.r_i}\big|_{1 \leq i \leq n}$$

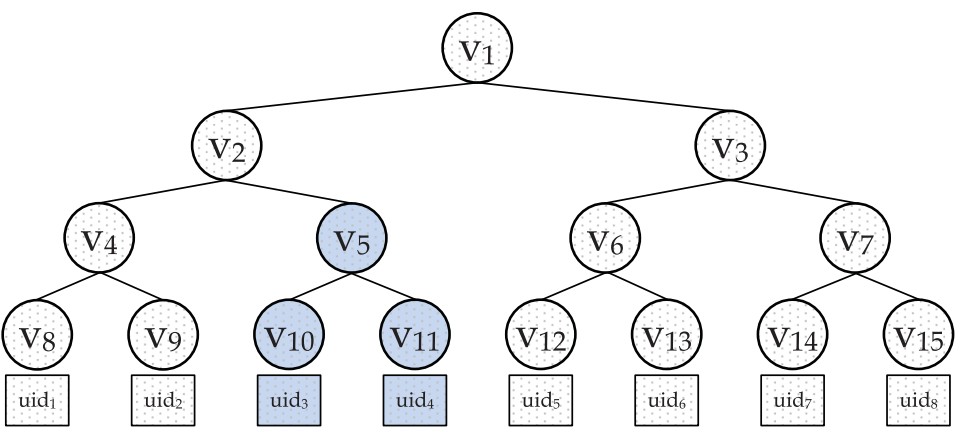

**Fig 3. A binary tree to manage attribute group members.**

*KEKGen(i, $k_i$, $v_j$, uid, mk, pp)→$KEK_i$*: The KEKGen algorithm is used to generate the key encryption key $KEK_i$ associated with an attribute *i*. To generate the $KEK_i$, the TA first creates an attribute group $G_i$ whose members are users bearing the attribute *i*. As in [24], the TA then establishes a binary tree to manage the members of $G_i$ as shown in Fig 3. The leaf nodes of the tree represent users. Each node in the tree holds a unique value $v_t \in \mathbb{Z}_p$. The path from the root node to a leaf node forms the path key *pkey* of a user. For example, the *pkey* for user $uid_5$ is $\{v_{12}, v_6, v_3, v_1\}$. Also, for each attribute group $G_i$, there is a set of minimum cover nodes $min(G_i)$. For instance, suppose the members of the attribute group $G_i$ are, $[uid_1, uid_2, uid_3, uid_4, uid_5, uid_6]$. The $min(G_i)$ for this list of members is $\{v_2, v_6\}$. As seen, there is an intersection $v_j$ between $min(G_i)$ and *pkey* for each member of $G_i$. For example the intersection $v_j$ for $uid_5$ is at node $v_6$. In addition, each attribute group $G_i$ is given a unique key $k_i \in_R \mathbb{Z}_p$. TA then computes attribute group information as $GI = k_i/v_j$, which is used during ciphertext generation. To generate a $KEK_i$ associated with a group $G_i$ for a user *uid*, the KEKGen algorithm computes $KEK_i$ as follows:

$$KEK_i = H(uid)^{r_i \cdot \frac{k_i}{v_j}}\big|_{1 \leq i \leq n, v_j \in pkey(uid) \cap min(G_i)}$$

Note that, this is computed for every attribute group the user belongs to.

3. **Encryption**

   The encryption phase consists of two sub-phases. The local encryption phase and the cloud encryption phase.

   *Local encryption*: The local encryption phase has one algorithm, the Encrypt algorithm which is executed by the DO. To encrypt data *M*, the DO first defines an *OBDD* access structure and uses the Encrypt algorithm to complete the local encryption.

   *Encrypt(OBDD, M, pp)→$CT_l$*: The Encrypt algorithm randomly chooses $s \in_R \mathbb{Z}_p$ and computes: $\tilde{C} = M.Y^s$, $C_0 = g^s$ and $C_1 = h_1^s$. The partial ciphertext $CT_l$ produced as output by the Encrypt algorithm is:

   $$CT_l = (OBDD, \tilde{C}, C_0, C_1).$$

   The $CT_l$ is then sent to the ESC for the cloud encryption and storage.

   *Cloud encryption*: The cloud encryption has one algorithm, the CldEncrypt algorithm

executed by the ESC. Upon receiving the $CT_l$ from the data owner, the ESC requests for attribute group information from TA for each attribute in the *OBDD* access structure. The TA sends $g^{GI} = g^{k_i/v_j}$ to the ESC. Using the CldEncrypt algorithm, the ESC then securely generates the complete ciphertext of the data $M$ by computing a header $C_{hdr}$ associated with each attribute in the access structure.

*CldEncrypt*$(CT_l, k_i, v_j, pp) \rightarrow CT$: The CldEncrypt algorithm computes the header as follows:

$$\forall i \in I : C_{hdr} = C_1.g^{GI} = h_1^s.g^{\frac{k_i}{v_j}}\Big|_{v_j \in min(G_i), k_i \in \mathbb{Z}_p}.$$

Where, $I$ is the attribute set of the OBDD access structure embedded in $CT_l$. After generating the headers associated with the ciphertext attributes, the ESC stores the ciphertext $CT$ as:

$$CT = (OBDD, \tilde{C}, C_0, C_1, C_{hdr})$$

Note that even without a dummy attribute, the ESC does not still obtain any information about the data $M$ during the header generation as it does not know the value of $s$.

4. **Decryption**

To minimize the high computation demand on the DUs, we propose an outsourced partial decryption of the data. Thus, the data decryption phase consists of the outsourced decryption and the local decryption sub-phases.

*Outsourced decryption*: The outsourced decryption phase is executed by the DC through the CldDecrypt algorithm. To decrypt the ciphertext $CT$, the DU first blinds his/her keys. The DU randomly chooses $x \in_R \mathbb{Z}_p$ and blinds the keys as:

$$(D_{i1})^x = g^{z_i.x}.H(uid)^{r_i.x}, (D_{i2})^x = H(uid)^{\alpha.r_i.x}\Big|_{1 \leq i \leq n}$$

$$(KEK_i)^x = H(uid)^{r_i.\frac{k_i}{v_j}.x}\Big|_{1 \leq i \leq n, v_j \in path(uid) \cap min(G_i)}$$

The DU then sends the blinded keys to the DC. The DU also requests the ESC to send $CT$ to DC. The ESC responds by sending the $C_0$ and $C_{hdr}$ parts of $CT$ to DC, and the $\tilde{C}$ part to the DU. Upon receiving the required $CT$ parts from the ESC, DC executes the CldDecrypt algorithm.

*CldDecrypt*$(CT, D_{i1}, D_{i2}, KEK_i, pp) \rightarrow C_{tkn}/\perp$: The CldDecrypt algorithm checks whether DU's attribute set satisfies the OBDD access structure in the ciphertext. If it does, it computes a token $C_{tkn}$ as:

$$
\begin{aligned}
C_{tkn} &= \prod_{i=1}^{n} \frac{e(D_{i1}, C_0).e(KEK_i, h_2)}{e(D_{i2}, C_{hdr})} \\[2mm]
&= \prod_{i=1}^{n} \frac{e(g^{z_i.x}.H(uid)^{r_i.x}, g^s).e(H(uid)^{r_i.\frac{k_i}{v_j}.x}, h_2)}{e(H(uid)^{\alpha.r_i.x}, h_1^s.g^{\frac{k_i}{v_j}})} \\[2mm]
&= \prod_{i=1}^{n} \frac{e(g,g)^{z_i.s.x}.e(H(uid),g)^{r_i.s.x}.e(H(uid),g)^{\alpha.r_i.\frac{k_i}{v_j}.x}}{e(H(uid),g)^{r_i.s.x}.e(H(uid),g)^{\alpha.r_i.\frac{k_i}{v_j}.x}} \\[2mm]
&= \prod_{i=1}^{n} e(g,g)^{z_i.s.x} = e(g,g)^{ysx}
\end{aligned}
$$

The generated $C_{tkn}$ is then sent to the DU. Otherwise, it returns $\perp$.

*Local decryption*: Upon receiving $C_{tkn}$ from DC and $\tilde{C}$ from the ESC, DU executes the Decrypt algorithm.

*Decrypt*$(C_{tkn}, CT, pp) \rightarrow (M)$: The Decrypt algorithm recovers $M$ as:

$$M = \frac{\tilde{C}}{(C_{tkn})^{1/x}} = \frac{MY^s}{(e(g,g)^{ysx})^{1/x}} = \frac{M.e(g,g)^{ys}}{e(g,g)^{ys}}.$$

5. **Revocation**

When a user is revoked of an attribute $i$, the TA updates the attribute group from $G_i$ to $G_i'$. For example, from Fig 3, if users $uid_3$ and $uid_4$ (the blue leaf nodes) are revoked of the attribute $i$, the new minimum cover node set $min(G_i')$ associated with the updated group $G_i'$ is $\{v_4, v_6\}$ which does not now intersect with $uid_3$ and $uid_4$'s *pkey*s. TA also chooses a new group key $k_i' \in \mathbb{Z}_p$ for $G_i'$. TA then executes the UpInfo algorithm to generate the update key $uk_i$ used for updating the ciphertext and the user keys.

*UpInfo*$(i, pp) \rightarrow uk_i$: The UpInfo algorithm computes the update key $uk_i$ as:

$$uk_{ic} = \frac{v_j.k_i'}{k_i.v_j'},$$

where $v_j' \in min(G_i')$ for updating the ciphertext and

$$uk_{ik} = \frac{k_i'}{v_j'} - \frac{k_i}{v_j},$$

where $v_j' \in path(uid) \cap min(G_i')$ for updating keys of non-revoked users. The TA updates the attribute group information to $GI'$ as:

$$GI' = GI \times uk_{ic} = \frac{k_i}{v_j} . \frac{v_j.k_i'}{k_i.v_j'} = \frac{k_i'}{v_j'}.$$

The TA then sends $g^{GI'} = g^{k_i'/v_j'}$ to the ESC to update the ciphertext and uses $uk_{ik}$ to update the keys of all the non-revoked DUs in the group.

*Ciphertext update*: Upon receiving the updated attribute group information, the ESC executes the CTUpdate algorithm to update the ciphertext.

*CTUpdate*$(CT, uk_i, i, pp) \rightarrow CT'$: The CTUpdate algorithm first randomly selects $s' \in_R \mathbb{Z}_p$ and updates $CT$ as:

$$
\begin{aligned}
CT' = \quad & (OBDD, \\
& \tilde{C} = M.Y^{(s+s')}, \\
& C_0 = g^{(s+s')}, \\
& C_1 = h_1^{(s+s')}, \\
\forall i = unrevoked: \quad & C_{hdr} = h_1^{(s+s')}.g^{\frac{k_i}{v_j}}, \\
\forall i = revoked: \quad & C_{hdr} = h_1^{(s+s')}.g^{\frac{k_i'}{v_j'}})
\end{aligned}
$$

Note that, for the revoked attribute, the ESC then uses the newly received $g^{GI'} = g^{k_i'/v_j'}$ and $h^{(s+s')}$ to compute the new header. ESC replaces $CT$ with $CT'$.

*Key update*: In this work, it is only the $KEK_i$ key that is updated. The $KEK_i$ is updated for

each non-revoked DU in the group by executing the KeyUpdate algorithm.

$KeyUpdate(i, uk_{ik}, KEK_i, pp) \rightarrow KEK_i'$: The KeyUpdate algorithm updates the $KEK_i$ associated with revoked attribute $i$ for each non-revoked DU to $KEK_i'$ as:

$$
\begin{aligned}
KEK_i' &= H(uid)^{r_i \cdot \frac{k_i}{v_j}}.H(uid)^{r_i \cdot \left( \frac{k_i'}{v_j'} - \frac{k_i}{v_j} \right)} \\
&= H(uid)^{r_i \cdot \frac{k_i'}{v_j}}
\end{aligned}
$$

## 6 Security analysis

In this section, we present a security proof of the CESCR scheme.

*Theorem 1: Suppose there is a PPT adversary $\mathcal{A}$ that can win our CPA security game with a non-negligible advantage $\varepsilon$, we can construct a simulator $\mathcal{B}$ that solves the DBDH problem with a non-negligible advantage.*

*Proof*: Let $\mathbb{G}$ and $\mathbb{G}_{\mathbb{T}}$ be two multiplicative cyclic groups of prime order $p$. Let $g$ be the generator of $\mathbb{G}$ and $e : \mathbb{G} \times \mathbb{G} \rightarrow \mathbb{G}_T$ be a bilinear map. The DBDH challenger $\mathcal{C}$ sends the tuple $(g, A = g^a, B = g^b, C = g^c, Z)$, where $a, b, c, z \in_R \mathbb{Z}_p$ to $\mathcal{B}$ and $\mathcal{B}$ is asked to output $v$. If $v = 0$, $Z = e(g, g)^{abc}$. Otherwise, $Z$ is a random value in $\mathbb{G}_{\mathbb{T}}$. $\mathcal{B}$ plays the role of the challenger in the CPA security game as follows:

**Initialization**: Adversary $\mathcal{A}$ declares a challenge access structure $\mathcal{OBDD}^*$ and attribute $i^*$ to $\mathcal{B}$.

**Setup**: $\mathcal{B}$ first sets $y = ab$. Then, $\mathcal{B}$ sets $h_1 = g^{1/\alpha}$, $h_2 = g^\alpha$, where $\alpha \in_R \mathbb{Z}_p$, and defines $Y = e(g, g)^y = e(g, g)^{ab} = e(A, B)$. $\mathcal{B}$ sends the public keys $pp = \{e, g, h_1, h_2, \mathbb{G}, Y\}$ to $\mathcal{A}$.

**Phase I**: $\mathcal{A}$ submits secret key and $KEK_i$ queries to $\mathcal{B}$. $\mathcal{A}$ requests the secret keys by submitting the attribute set $S$ belonging to a user $uid$ to $\mathcal{B}$. $S$ satisfies $\mathcal{OBDD}^*$ but $i^*$ is revoked. $\mathcal{B}$ creates a list $HL$: $<uid, H>$ and a table $T :< uid, S, KEK_i, \mathcal{D}_{i1}, \mathcal{D}_{i2} >$ which are initially empty. $\mathcal{B}$ checks the $HL$ to confirm whether the pair $<uid, H>$ exists and does the following:

1. If the pair $<uid, H>$ exists, $\mathcal{B}$ responds by sending $H$ which is the hash value associated with $uid$ to $\mathcal{A}$.

2. Otherwise, $\mathcal{B}$ generates $H$ for $uid$ as follows:

$$
\forall uid : H = g^u.
$$

Where $u \in_R \mathbb{Z}_p$.

3. $\mathcal{B}$ stores the pair $<uid, H>$ in $HL$ and sends $H$ to $\mathcal{A}$. Note that, $\mathcal{A}$ can query for $H$ at any time and $\mathcal{B}$ responds as the same.

Then, $\mathcal{B}$ checks $T$ to confirm whether the tuple $< uid, S, KEK_i, \mathcal{D}_{i1}, \mathcal{D}_{i2} >$ exists. If it exists, $\mathcal{B}$ sends the associated $KEK_i$ and $(\mathcal{D}_{i1}, \mathcal{D}_{i2})$ to $\mathcal{A}$. Otherwise, $\mathcal{B}$ does the following:

1. First, $\mathcal{B}$ checks $HL$ for the hash value associated with $uid$. If it exists, $\mathcal{B}$ extracts it for usage during key generations. Else, $\mathcal{B}$ uses the above hash generation steps to generate $H$ for $uid$. Then, for each $i \in S$, $\mathcal{B}$ randomly chooses $s_i', r_i \in \mathbb{Z}_p$ and sets $z_i = s_i'$, where $n = |S|$ and $z_n = y - \sum_{i=1}^{n-1} z_i = ab - \sum_{i=1}^{n-1} s_i'$. Then, $\mathcal{B}$ uses the $(\mathcal{D}_{i1}, \mathcal{D}_{i2}) \leftarrow \text{KeyGen}(S, uid, mk, pp)$ algorithm to generate the secret key

$$
\begin{aligned}
\mathcal{D}_{i1} &= g^{z_i}.H(uid)^{r_i} = g^{s_i'}.g^{ur_i}, \\
\mathcal{D}_{i2} &= H(uid)^{\alpha.r_i} = g^{u.\alpha.r_i}|_{1 \le i \le n}.
\end{aligned}
$$

2. $\mathcal{B}$ then randomly chooses $k_i'' \in_R \mathbb{Z}_p$ and minimum cover node $v_j'' \in \mathbb{Z}_p$ for each $i \in S \wedge i \neq i^*$. $\mathcal{B}$ also randomly chooses $v_{i^*}'' \in_R \mathbb{Z}_p$ and $k_{i^*}'' \in_R \mathbb{Z}_p$ as the minimum cover node and group key for $i^*$, respectively. It then sets attribute group key $k_i$ as follows:

$$\forall i = i^* : k_i = k_{i^*}''$$
$$\forall i \neq i^* : k_i = bk_i''$$

$\mathcal{B}$ then uses the $KEK_i \leftarrow \text{KEKGen}(i, k_i, v_j, uid, mk, pp)$ algorithm to generate the key encryption key $KEK_i$ for each attribute as:

$$\forall i = i^* : KEK_{i^*} = H(uid)^{r_i \cdot \frac{k_{i^*}}{v_{i^*}}} = g^{\frac{u \cdot r_i \cdot k_i''}{v_i''}}$$
$$\forall i \neq i^* : KEK_i = H(uid)^{r_i \frac{k_i}{v_j}} = B^{\frac{u \cdot r_i \cdot k_i''}{v_j''}}$$

3. $\mathcal{B}$ adds the $KEK_i$ and $(\mathcal{D}_{i1}, \mathcal{D}_{i2})$ in a tuple $< uid, S, KEK_i, \mathcal{D}_{i1}, \mathcal{D}_{i2} >$ and stores it in the table $T$. $\mathcal{B}$ sends the $\mathcal{D}_{i1}, \mathcal{D}_{i2}$ and $KEK_i$ values to $\mathcal{A}$.

$\mathcal{A}$ may decide to ask for an update key for another revoked attribute $i \neq i^*$, $\mathcal{B}$ randomly chooses $\bar{k}_i, \bar{v}_j \in \mathbb{Z}_p$ and using $uk_{i \neq i^*} \leftarrow \text{UpInfo}(i, pp)$ algorithm, it generates an update key $uk_i = \left( \frac{\bar{k}_i}{\bar{v}_j} - \frac{k_i''}{v_j''} \right)$. $\mathcal{B}$ then computes a new $KEK'$ using the $KeyUpdate$ algorithm and submits it to $\mathcal{A}$.

**Challenge**: Once adversary $\mathcal{A}$ decides Phase 1 is over, it submits two messages $M_0$ and $M_1$ of equal length to $\mathcal{B}$ and set the access structure as $\mathcal{OBDD}^*$ and $i^*$ as a revoked attribute. $\mathcal{B}$ randomly flips a coin $\mu \in \{0, 1\}$ and encrypts $M_\mu$ as $CT_l$ using the $CT_l \leftarrow \text{Encrypt}(\mathcal{OBDD}^*, M_\mu, pp)$ algorithm. $CT_l$ is generated as: $\tilde{C} = M_\mu \cdot e(g, g)^{yc} = M_\mu \cdot e(g, g)^{abc}$, $C_0 = g^c = C$ and $C_1 = h_1^c = g^{c/\alpha} = C^{1/\alpha}$.

$$CT_l = < \mathcal{OBDD}^*, \tilde{C}, C_0, C_1 >$$

Then, for each $i \in I^*$, $I^*$ is the set of attributes in $\mathcal{OBDD}^*$, $\mathcal{B}$ generates group attribute information as:

$$\forall i = i^* : g^{k_{i^*}''/v_{i^*}''}$$
$$\forall i \neq i^* : B^{k_i''/v_j''}$$

$\mathcal{B}$ then generates headers associated with the ciphertext attributes using the $CT \leftarrow \text{CldEncrypt}(CT_l, k_i, v_j, pp)$ algorithm as:

$$\forall i = i^* : C_{hdr} = C^{1/\alpha} \cdot g^{k_{i^*}''/v_{i^*}''}$$
$$\forall i \neq i^* : C_{hdr} = C^{1/\alpha} \cdot B^{k_i''/v_j''}$$

The generated $CT$ is:

$$CT = < \mathcal{OBDD}^*, \tilde{C}, C_0, C_1, C_{hdr} >$$

$\mathcal{B}$ then updates the ciphertext using the $CT' \leftarrow \text{CTUpdate}(CT, uk_{i^*}, i^*, pp)$ algorithm. $\mathcal{B}$ first

randomly chooses $s' \in_R \mathbb{Z}_p$ and updates the ciphertext as:

$$
\begin{aligned}
CT' = \quad & (OBDD*, \\
& \tilde{C} = M_\mu.Y^{(c+s')} = M_\mu.Z.e(g,g)^{abs'}, \\
& C_0 = g^{(c+s')}, \\
& C_1 = h_1^{(c+s')}, \\
\forall i = i^* : \quad & C_{hdr} = h_1^{(c+s')}.g^{\frac{k^*}{v^*}}, \\
\forall i \neq i^* : \quad & C_{hdr} = h_1^{(c+s'')}.B^{\frac{k_i''}{v_j'}})
\end{aligned}
$$

For $i^*$, $\mathcal{B}$ generates $g^{\frac{k^*}{v^*}}$, where $k^*, v^* \in_R \mathbb{Z}_p$ and uses it together with the updated $C_1$ to generate the $C_{hdr}$. $\mathcal{B}$ sets $CT'$ as the challenger ciphertext $CT^*$ and sends it to $\mathcal{A}$.

**Phase II**: $\mathcal{A}$ continues to adaptively submit key queries as in phase I.

**Guess**: Adversary $\mathcal{A}$ then outputs a guess $\mu'$ for $\mu$. If $\mu' = \mu$, $\mathcal{B}$ outputs $v' = 0$, i.e., $Z = e(g, g)^{abc}$. Otherwise, $\mathcal{B}$ outputs $v' = 1$, i.e., $Z$ is a random number in $\mathbb{G}_T$.

In the case $v = 1$, the adversary gains no information about $M_\mu$. Thus, $Pr[\mu' \neq \mu | v = 1] = \frac{1}{2}$. $\mathcal{B}$ randomly guesses $v'$ for $v$ when $\mu' \neq \mu$ with a probability $Pr[v' = v | v = 1] = \frac{1}{2}$.

If $v = 0$, the adversary sees encryption of the message $M_\mu$. By definition, the advantage of the adversary in this situation is $\varepsilon$. Thus, $Pr[v' = v | v = 0] = \frac{1}{2} + \varepsilon$.

Therefore, the overall advantage of $\mathcal{B}$ in winning the above game is:

$$
\begin{aligned}
&= \frac{1}{2}.(Pr[v' = v | v = 0]) + \frac{1}{2}.(Pr[v' = v | v = 1]) - \frac{1}{2} \\
&= \frac{1}{2}.\left(\frac{1}{2} + \varepsilon\right) + \frac{1}{2}.\frac{1}{2} - \frac{1}{2} = \frac{\varepsilon}{2}
\end{aligned}
$$

# 7 Simulation and performance analysis

## 7.1 Performance analysis

In this section, we analyze and compare our scheme with CP-ABE schemes in [17, 18, 24, 25, 49] in terms of revocation, boundedness, expressiveness and efficiency features. As shown in Table 1, all the schemes including ours are built using the prime order groups except the Zhang et al.'s scheme [49] which uses the composite order group. The schemes [17, 18] and ours are unrestricted and more expressive as they are based-on the non-monotonic and non-restrictive OBDD access structure. Meanwhile, the [18, 24] schemes which are based on the access tree access structure and the [49] scheme which is based on the LSSS access structure are less expressive. Our scheme and the Li et al.'s scheme [25] partially outsource their encryption and decryption tasks and thus, they are computationally more efficient on the data owner and user side. The computation tasks during encryption and decryption in the rest of the schemes are entirely performed by the data owners and data users and hence computationally more demanding on the data owner and data user sides. All the CP-ABE schemes except the [24] scheme are collusion resistant. Immediate attribute/user revocation is achieved in [24, 25] and our schemes, meanwhile, the rest of the schemes do not include an attribute/user revocation mechanism. Only the [17, 18] schemes are bounded, the rest of the CP-ABE schemes including ours are unbounded.

In the same Table 1, we present the storage comparison of the CESCR scheme in relation to the other CP-ABE schemes. We use $|k|$ to denote the number of user attributes, $|l|$ to denote

**Table 1. Feature and storage comparison of CP-ABE schemes.**

| | Schemes | | | | | |
|---|---|---|---|---|---|---|
| | [49] | [24] | [17] | [25] | [18] | CESCR |
| Key size | $|k| + 2$ | $2|k| + 1 + |pk|$ | 2 | $3|k| + 6$ | 2 | $3|k|$ |
| Ciphertext size | $3|l| + 4 + |A|$ | $3|l| + 2 + |A|$ | $2 + |R| + |A|$ | $2|l| + 7 + |A|$ | $2 + |R| + |A|$ | $|l| + 3 + |A|$ |
| Unbounded | ✓ | ✓ | × | ✓ | × | ✓ |
| Revocation | × | ✓ | × | ✓ | × | ✓ |
| Coll-Resist | ✓ | × | ✓ | ✓ | ✓ | ✓ |
| Encryption | DO | DO | DO | Par-out | DO | Par-out |
| Decryption | DU | DU | DU | Par-out | DU | Par-out |
| Expressiveness | LSSS | Access tree | OBDD | Access tree | OBDD | OBDD |
| Group Order | Composite | Prime | Prime | Prime | Prime | Prime |

*$|pk|$ is path key size, Coll-Resist is collusion resistance, Par-out is partially outsourced, DO is data owner, DU is data user.

the number of attributes in the ciphertext, $|A|$ to denote the size of the access structure and $|R|$ is the number of routes that satisfy an OBDD access structure. Note that the same attribute can be repeated across multiple routes that satisfy the OBDD access structure. It can be observed that the CESCR scheme has optimal ciphertext storage efficiency only bettered by the [17, 18] schemes. This is because the only attribute element included in the CESCR's ciphertext is the one associated with the attribute group keys. However, the CESCR scheme performs slightly worse than the other schemes except the Li *et al.*'s scheme [25] in key storage. This is because all the key components are interlinked for each attribute, which helps in preventing collusion attacks.

The computational comparisons are presented in Table 2. The comparison is done in terms of encryption, decryption and key generation costs. The encryption and decryption costs are analyzed on both the data owner and cloud sides. Here, we use $|d|$ to denote the number of attributes involved in satisfying an access structure or simply the number of attributes involved

**Table 2. Computation comparison of CP-ABE schemes.**

| | | | Schemes | | | | | |
|---|---|---|---|---|---|---|---|---|
| | | | [49] | [24] | [17] | [25] | [18] | CESCR |
| Encryption Cost | Mult | DO | $11|l| + 2$ | 1 | $|l|$ | 2 | $|l|$ | 1 |
| | | Cloud | n/a | $|l|$ | n/a | n/a | n/a | $|l|$ |
| | Expo | DO | $7|l| + 4$ | $2|l| + 2$ | $|l| + 2$ | 6 | $|l| + 2$ | 3 |
| | | Cloud | n/a | $|l|$ | n/a | $2|l|$ | n/a | n/a |
| | Pair | DO | n/a | n/a | n/a | 2 | n/a | n/a |
| Decryption Cost | Mult | DU | $4|d| - 2$ | $\geq |d| + 2$ | 2 | 4 | 2 | 1 |
| | | Cloud | n/a | n/a | n/a | $\geq |d| + 2$ | n/a | $2|d|$ |
| | Expo | DU | $3|d| + 1$ | 1 | n/a | 4 | n/a | 4 |
| | Pair | DU | $2|d| + 3$ | $\geq 2|d| + 1$ | 2 | n/a | 2 | n/a |
| | | Cloud | n/a | n/a | n/a | $\geq 2|d| + 4$ | n/a | $3|d|$ |
| Key Gen Cost | Mult | | $2|k| + 4$ | $|k| + 1$ | 1 | $4|k| + 10$ | 1 | $|k|$ |
| | Expo | | $2|k| + 3$ | $3|k| + 1$ | 2 | $4|k| + 6$ | 2 | $4|k|$ |
| | Pair | | n/a | n/a | n/a | 1 | n/a | n/a |

*Multi, Expo and Pair represent the multiplication, exponentiation and pairing operations, respectively. DO is data owner and DU is data user.

in decryption. The [25] scheme and our scheme outsource the attribute operations during encryption and decryption to the cloud. For the rest of the schemes, the encryption and decryption tasks are entirely performed by the data owner and data user, respectively. Thus, on the DO side, the CESCR scheme has the least computation demand during encryption, as it requires only one multiplication and three exponentiation operations which are independent of the number of attributes in the ciphertext. Zhang *et al.*'s [49] scheme is the most demanding on the DO side computationally. Unlike the scheme [25] which performs 2 pairing and $2|l|$ exponentiation operations in the cloud during encryption, our scheme only performs $|l|$ multiplications, which also makes it more efficient on the cloud side during encryption. Similarly, during decryption, our scheme is computationally the least demanding on the DU side as it requires only one multiplication and four exponentiation operations and the Zhang *et al.*'s [49] scheme is still the most demanding. However, on the cloud side during decryption, our scheme is slightly bettered by the Li *et al.*'s [25] scheme, this is because our scheme requires more pairing operations. In key generation, though our scheme is computationally more demanding due to its linking of all the key components for all the user attributes, it still performs better than the Li *et al.*'s scheme [25].

## 7.2 Experimental analysis

To explicitly demonstrate the efficiency of the CESCR scheme, we simulated the scheme in comparison with the [25, 24] schemes which we refer to in the experiment as the "LZQH scheme" and "H-N scheme", respectively. The implementation was done using the Charm crypto framework [53]. We used the "SS512" curve which is a super-singular symmetric elliptic curve over 512-bit base field having a 160-bit curve group order. The experiment was carried out on a desktop computer with a 3.20GHz processor and 4.0 GB RAM running the Ubuntu 12.04 operating system. Each experiment was repeated 20 times, and we averaged the results and are shown in Fig 4.

Fig 4(a) shows the setup computation time against the size of the attribute universe. It can be observed that all the schemes have constant computation time against the number of attributes. The schemes are all unbounded schemes and thus the number of parameters generated at setup does not depend on the size of the attribute universe. Our scheme generates more parameters and thus takes more computation time at setup as compared to the LZQH and H-N schemes. The LZQH scheme generates the least number of parameters during setup and hence the low computation time.

In Fig 4(b), we show the variation of computation time against the number of user attributes during key generation. Our scheme outperforms the LZQH scheme because of its fewer key elements. However, the H-N scheme exhibits the best performance during key generation because of its low exponentiation operation requirements.

Fig 4(c) and 4(d) show the variation of computation time in local and cloud encryptions against the number of attributes in the ciphertext. For our scheme and the LZQH scheme, since they both outsource their attribute operations to the Cloud, the computation time is constant against the varying ciphertext attribute number during the local encryption. For the H-N scheme, the computation time during local encryption increases with the increase in the number of attributes in the ciphertext. However, the computation time increases with the increasing ciphertext attribute number during the cloud encryption for all the schemes. In both cases, our scheme generally performs better than the LZQH and H-N schemes because of having fewer elements and exponentiation operations to be computed by the Cloud. Also, during local encryption, unlike the LZQH scheme, our scheme does not perform any pairing operations and there are no operations associated with a dummy attribute as in the LZQH scheme.

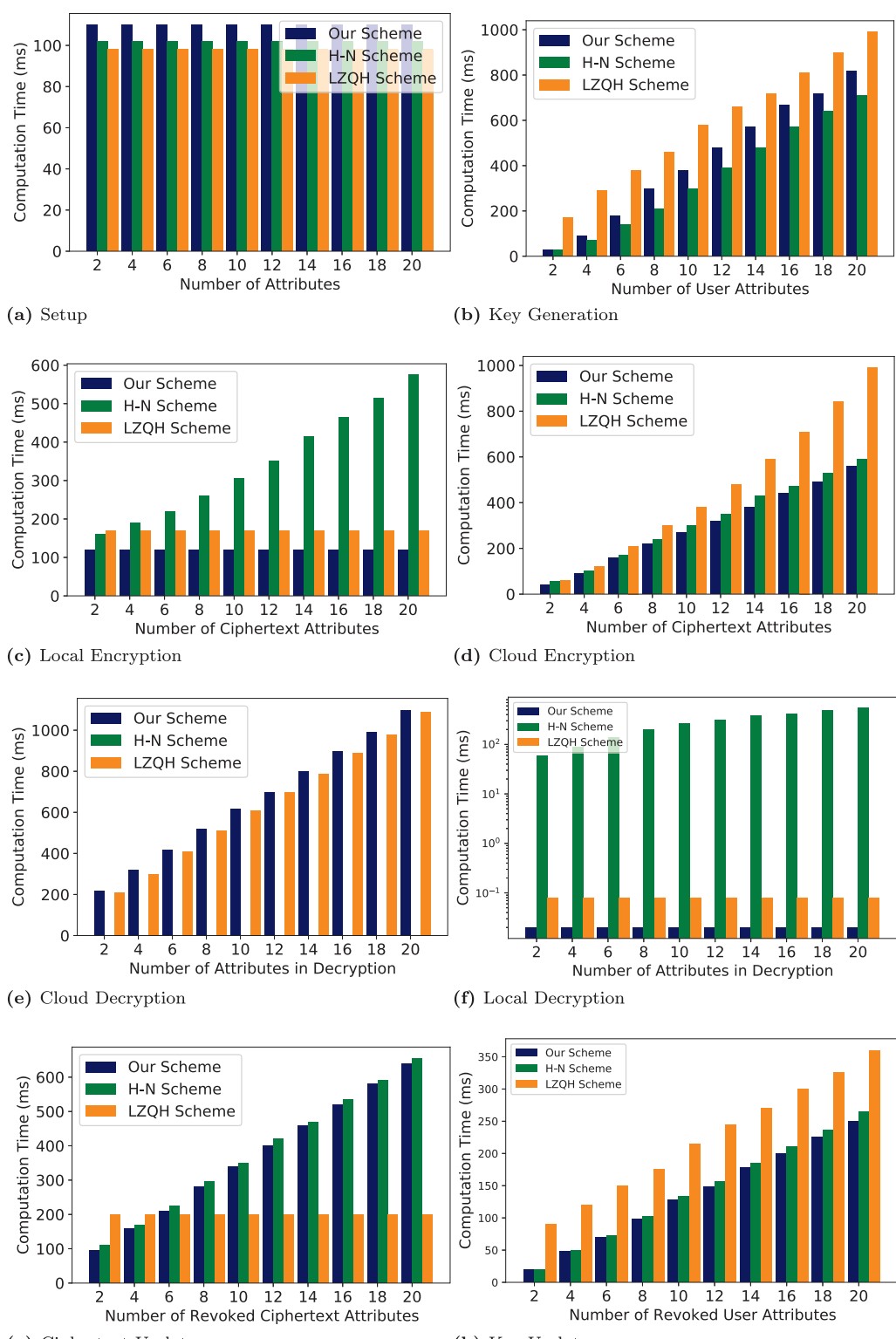

**Fig 4. Simulation results of the CESCR scheme in comparison with the LZQH [25] and H-N [24] schemes.**

We show the decryption computation times against the varying number of attributes involved during decryption in Fig 4(e) and 4(f). For the cloud decryption, the computation time increases with an increase in the number of attributes involved in decryption for all the schemes except the H-N scheme that does not perform cloud decryption. Meanwhile, all the schemes except the H-N scheme exhibit constant computation times during the local decryption which are 0.02 ms and 0.08 ms for our scheme and the LZQH scheme, respectively. For local decryption, our scheme performs fewer multiplication and exponentiation operations as compared to the LZQH scheme, and thus the low computation time. All the attribute operations associated with decryption are performed by the user for the H-N scheme and hence the increase in computation time against the increase in number of attributes involved in decryption. In the cloud decryption, the difference in computation time between our scheme and the LZQH scheme is minimal.

In Fig 4(g) and 4(h), we show the variation of computation time for ciphertext update and key update against the number of revoked ciphertext and user attributes, respectively. For the ciphertext update, the computation time for our scheme and the H-N scheme increase with the increase in the number of revoked ciphertext attributes but remains constant for the LZQH scheme. This is because our scheme and the H-N scheme update the attribute elements associated with the revoked attributes. Meanwhile, in the LZQH scheme, only two ciphertext elements not related to the revoked attributes get updated and thus the constant computation time. However, unlike the H-N scheme that independently encrypts the header message, our scheme achieves better performance. For the key update, the computation time increases with the increase in the number of revoked user attributes for all the schemes. However, our scheme performs better, since it has fewer key elements that get updated as compared to the LZQH scheme and there is no independent decryption of group keys as compared to the H-N scheme.

In general, the proposed CESCR scheme is more expressive as it can handle the non-monotonic access policies without restrictions and is more efficient on the data user and data owner sides.

## 8 Conclusion

In this work, we focused on addressing data privacy and security concerns in collaborative ehealth systems. We proposed the CESCR scheme, which is a CP-ABE scheme whose main ingredients are, immediate attribute/user revocation, unboundedness, expressiveness, efficiency, and collusion resistance. We adapted the attribute group approach to address the immediate attribute/user revocation issues and bind the keys to the user identities to prevent collusion between data users. OBDD access structure was used to achieve expressivessness. A novel technique that limits the attribute elements in the ciphertext to only those associated with attribute group keys was proposed to achieve unboundedness and improved efficiency. The CESCR scheme further securely outsources the computationally demanding attribute operations in both encryption and decryption to the cloud without requiring a dummy attribute. We performed extensive security and performance analysis of the scheme in comparison with related CP-ABE schemes and the results show that the CESCR scheme is expressive, unbounded, secure, and efficient in comparison with the related CP-ABE schemes. The addition of traceability through the use of blockchain technology and policy hiding are interesting future considerations.

## Author Contributions

**Conceptualization:** Kennedy Edemacu.

**Data curation:** Kennedy Edemacu.

**Formal analysis:** Kennedy Edemacu.

**Funding acquisition:** Jong Wook Kim.

**Methodology:** Kennedy Edemacu.

**Project administration:** Jong Wook Kim.

**Resources:** Jong Wook Kim.

**Software:** Jong Wook Kim.

**Supervision:** Beakcheol Jang, Jong Wook Kim.

**Validation:** Kennedy Edemacu.

**Visualization:** Kennedy Edemacu, Beakcheol Jang.

**Writing – original draft:** Kennedy Edemacu.

**Writing – review & editing:** Beakcheol Jang, Jong Wook Kim.

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
