## [Decision Letter · Decision Letter 0]

1 Mar 2021

PONE-D-21-04536

CESCR: CP-ABE for Efficient and Secure Sharing of Data in Collaborative eHealth with Revocation and no Dummy Attribute

PLOS ONE

Dear Dr. Kim,

Thank you for submitting your manuscript to PLOS ONE. After careful consideration, we feel that it has merit but does not fully meet PLOS ONE’s publication criteria as it currently stands. Therefore, we invite you to submit a revised version of the manuscript that addresses the points raised during the review process.

We look forward to receiving your revised manuscript.

Kind regards,

Pandi Vijayakumar, Ph.D

Academic Editor

PLOS ONE

Journal Requirements:

Reviewers' comments:

Reviewer's Responses to Questions

**Comments to the Author**

1. Is the manuscript technically sound, and do the data support the conclusions?

Reviewer #1: Partly

Reviewer #2: Yes

2. Has the statistical analysis been performed appropriately and rigorously? 

Reviewer #1: Yes

Reviewer #2: Yes

3. Have the authors made all data underlying the findings in their manuscript fully available?

Reviewer #1: Yes

Reviewer #2: Yes

4. Is the manuscript presented in an intelligible fashion and written in standard English?

Reviewer #1: Yes

Reviewer #2: Yes

5. Review Comments to the Author

Reviewer #1: In this paper, the authors propose the CESCR solution to solve the problem of instant attribute/user cancellation and collusion, and achieve unboundedness and expressiveness.

Although this scheme implements attribute/user revocation, there are other problems. To accept it, there is still a long way to go. We point out the specific problems:

1. The paper lacks innovative points. The fifth point of user anti-collusion is actually something that has already been mentioned in the first point, and the attribute/user cancellation mentioned by the author is only the technology of other solutions. The fourth point of innovation is also a feature of the OBDD access structure itself, rather than unique to the solution.

2. In the section of Security Analysis, the probability of challenger B's success has not been analyzed in detail. Please use the mathematical formula to analyze concretely.

3. In the section of Simulation and Performance Analysis, the authors only chose to compare with one plan, which makes the degree of persuasion was not strong.

4. The content format and reference format of this article are not accurate enough. Please double check and correct.

Reviewer #2: 1. The introduction part is well written.

2. The technical novelty of the paper is good. The research problem and research methods are described clearly. The paper is potentially worthy of publication.

3. The related work section is comprehensive. However, the authors are requested to analyse the following papers in the releted work section.

* An efficient anonymous authentication and confidentiality preservation schemes for secure communications in wireless body area networks, wireless networks.

* Efficient and secure anonymous authentication with location privacy for IoT-based WBANs, IEEE Transactions on Industrial Informatics.

4. The contribution of this paper is well.

6. PLOS authors have the option to publish the peer review history of their article (what does this mean?). If published, this will include your full peer review and any attached files.

Reviewer #1: No

Reviewer #2: No

---

## [Author Response · Author response to Decision Letter 0]

4 Apr 2021

Our responses to the reviewers' comments are included in the 'Response to Reviewers' file.

---

## [Decision Letter · Decision Letter 1]

19 Apr 2021

CESCR: CP-ABE for Efficient and Secure Sharing of Data in Collaborative eHealth with Revocation and no Dummy Attribute

PONE-D-21-04536R1

Dear Dr. Kim,

We’re pleased to inform you that your manuscript has been judged scientifically suitable for publication and will be formally accepted for publication once it meets all outstanding technical requirements.

Kind regards,

Pandi Vijayakumar, Ph.D

Academic Editor

PLOS ONE

Additional Editor Comments (optional):

The paper can be accepted in the present format.

Reviewers' comments:

Reviewer's Responses to Questions

**Comments to the Author**

1. If the authors have adequately addressed your comments raised in a previous round of review and you feel that this manuscript is now acceptable for publication, you may indicate that here to bypass the “Comments to the Author” section, enter your conflict of interest statement in the “Confidential to Editor” section, and submit your "Accept" recommendation.

Reviewer #2: All comments have been addressed

2. Is the manuscript technically sound, and do the data support the conclusions?

Reviewer #2: Yes

3. Has the statistical analysis been performed appropriately and rigorously? 

Reviewer #2: Yes

4. Have the authors made all data underlying the findings in their manuscript fully available?

Reviewer #2: Yes

5. Is the manuscript presented in an intelligible fashion and written in standard English?

Reviewer #2: (No Response)

6. Review Comments to the Author

Reviewer #2: (No Response)

7. PLOS authors have the option to publish the peer review history of their article (what does this mean?). If published, this will include your full peer review and any attached files.

Reviewer #2: No

---

## [Editor Report · Acceptance letter]

23 Apr 2021

PONE-D-21-04536R1 

CESCR: CP-ABE for Efficient and Secure Sharing of Data in Collaborative eHealth with Revocation and no Dummy Attribute 

Dear Dr. Kim:

I'm pleased to inform you that your manuscript has been deemed suitable for publication in PLOS ONE. Congratulations! Your manuscript is now with our production department. 

Kind regards, 

on behalf of

Dr. Pandi Vijayakumar 

Academic Editor

PLOS ONE